# In Vitro Development of Local Antiviral Formulations with Potent Virucidal Activity Against SARS-CoV-2 and Influenza Viruses

**DOI:** 10.3390/pharmaceutics17030349

**Published:** 2025-03-08

**Authors:** Juthaporn Ponphaiboon, Wantanwa Krongrawa, Sontaya Limmatvapirat, Sukannika Tubtimsri, Akanitt Jittmittraphap, Pornsawan Leaungwutiwong, Chulabhorn Mahidol, Somsak Ruchirawat, Prasat Kittakoop, Chutima Limmatvapirat

**Affiliations:** 1Department of Industrial Pharmacy, Faculty of Pharmacy, Silpakorn University, Nakhon Pathom 73000, Thailand; augusto_sc@hotmail.co.th (J.P.); krongrawa_w@su.ac.th (W.K.); limmatvapirat_s@su.ac.th (S.L.); 2Natural Products Research Center (NPRC), Faculty of Pharmacy, Silpakorn University, Nakhon Pathom 73000, Thailand; 3Pharmaceutical Intellectual Center “Prachote Plengwittaya”, Faculty of Pharmacy, Silpakorn University, Nakhon Pathom 73000, Thailand; 4Department of Pharmaceutical Technology, Faculty of Pharmaceutical Sciences, Burapha University, Chonburi 20131, Thailand; sukannika@go.buu.ac.th; 5Department of Microbiology and Immunology, Faculty of Tropical Medicine, Mahidol University, Bangkok 10400, Thailand; akanitt.jit@mahidol.ac.th (A.J.); pornsawan.lea@mahidol.ac.th (P.L.); 6Laboratory of Natural Products and Medicinal Chemistry, Chulabhorn Research Institute, Bangkok 10210, Thailandsomsak@cri.or.th (S.R.); prasat@cri.or.th (P.K.); 7Program in Chemical Sciences, Chulabhorn Graduate Institute, Bangkok 10210, Thailand; 8Center of Excellence on Environmental Health and Toxicology (EHT), OPS, Ministry of Higher Education, Science, Research and Innovation, Bangkok 10400, Thailand

**Keywords:** D-limonene, monolaurin, cetylpyridinium chloride, SARS-CoV-2, influenza virus, virucidal activity, prevention, therapeutic applications

## Abstract

**Background/Object:** This study investigates the in vitro antiviral potential of D-limonene (DLM), monolaurin (ML), and cetylpyridinium chloride (CPC) in formulations targeting SARS-CoV-2 and influenza viruses. The aim was to develop oral and nasal formulations with optimized concentrations of these active ingredients to evaluate their efficacy, safety, and stability. **Methods:** Oral (formulation D) and nasal (formulation E) products were developed using specific concentrations of DLM (0.2–0.3% *w*/*w*), ML (0.1–0.2% *w*/*w*), and CPC (0.05–0.075% *w*/*w*). In vitro virucidal activity assays were conducted to assess the antiviral efficacy of the formulations against SARS-CoV-2 and influenza viruses. Stability testing was also performed under various storage conditions. **Results:** Formulation D (0.3% *w*/*w* DLM, 0.2% *w*/*w* ML, 0.05% *w*/*w* CPC, and 1.5% *w*/*w* Cremophor RH40) demonstrated a 3.875 ± 0.1021 log reduction and 99.99 ± 0.0032% efficacy against SARS-CoV-2 within 120 s. Formulation E (0.2% *w*/*w* DLM, 0.05% *w*/*w* CPC, and 0.75% *w*/*w* Cremophor RH40) showed a 2.9063 ± 0.1197 log reduction and 99.87 ± 0.0369% efficacy against SARS-CoV-2. Both formulations achieved >99.99% efficacy and log reductions exceeding 4.000 against various influenza strains. Stability testing confirmed optimal performance at 4 °C with no microbial contamination. **Conclusions:** The findings suggest that both formulations exhibit broad-spectrum antiviral activity against SARS-CoV-2 and influenza viruses in vitro. These results support their potential for further clinical evaluations and therapeutic applications, particularly in oral and nasal spray formulations.

## 1. Introduction

Coronavirus disease 2019 (COVID-19), caused by SARS-CoV-2, primarily infects lung epithelial cells via angiotensin-converting enzyme 2 in type II alveolar cells [1]. Despite lower expression in the nasopharynx and oral epithelium, active replication occurs in the upper respiratory tract, facilitating transmission through saliva and nasal secretions [2]. Influenza viruses A and B, responsible for seasonal outbreaks, target the respiratory tract and range from mild to severe illnesses, particularly in vulnerable populations [3].

D-limonene (DLM), a citrus-derived monoterpene, has shown antiviral activity by disrupting viral replication and enhancing immune responses, with proven efficacy against H1N1 and other viruses [4,5]. Classified as Generally Recognized as Safe (GRAS), DLM offers a safe antiviral option [6]. Monolaurin (ML), derived from lauric acid, disrupts viral membranes, impairing infection and replication, with broad-spectrum activity against respiratory pathogens [7,8]. GRAS-approved ML is being studied for incorporation into antiviral formulations [9]. Cetylpyridinium chloride (CPC), a quaternary ammonium compound, disrupts viral membranes, reducing SARS-CoV-2 and influenza viral loads [10,11]. It is GRAS-approved and used in oral antiseptics to prevent transmission [12].

Oral antiseptics like CPC, DLM, and ML effectively reduce SARS-CoV-2 and influenza viral loads in the oral and nasal cavities. CPC, alone or combined with chlorhexidine or DLM, shows potent virucidal activity without resistance [11,13,14,15,16]. ML inhibits SARS-CoV-2 and H1N1 with low cytotoxicity, enhancing immune defense [7,8]. Essential oils and CPC offer safe, short-term antiviral potential, but further studies are needed to optimize formulations for maximal efficacy [17,18].

This study aims to develop and assess oral and nasal formulations incorporating CPC (0.05–0.075% *w*/*w*), DLM (0.2–0.3% *w*/*w*), and ML (0.1–0.2% *w*/*w*) [7,16,19], focusing on their in vitro virucidal efficacy against SARS-CoV-2 and influenza viruses.

## 2. Materials and Methods

### 2.1. Materials

DLM and ML were sourced from TCI AMERICA (Tokyo Chemical Industry Co., Ltd., Kita-ku, Tokyo, Japan) and Shanghai Terppon Chemical Co., Ltd. (Zhao Jia Bang Road, Shanghai, China). CPC, surfactants, and menthol came from MySkin Recipes (Chanjao Longevity Co., Ltd., Bang Khen, Bangkok, Thailand) and Chemecosmetics Co., Ltd. (Prawet, Bangkok, Thailand). Sigma-Aldrich (St. Louis, MO, USA) supplied reagents, solvents, and microbial media. Peppermint oil and tartrazine were obtained from Chemipan and Adinop Co., Ltd. (Bang Khae, Bangkok, Thailand). Additional materials included nitric acid (Merck KGaA, Darmstadt, Hessen, Germany) and ICP standards (Agilent Technologies, Santa Clara, CA, USA). Antiviral testing utilized MEM, FBS, antibiotics, and GlutaMAX from Thermo Fisher Scientific (Life Technologies, Waltham, MA, USA).

### 2.2. Investigation of Optimal Surfactants for Formulations Containing DLM

Various surfactants (e.g., Tween 20, Tween 60, Tween 80, Cremophor RH40, Cremophor RH60, coco glucoside, decyl glucoside, Poloxamer 184, and Poloxamer 407) were evaluated in 1% *w*/*w* DLM formulations. Surfactants and DLM were dissolved in water at 62–65 °C and homogenized at 3800 rpm for 5 min (T25 digital Ultra-Turrax, IKA, Staufen im Breisgau, Baden-Württemberg, Germany). For Poloxamer 407, it was first swollen in cold water before being mixed with limonene. The formulations were then cooled, stored in amber bottles, and analyzed for % transmittance and pH after preparation. A temperature cycling test (alternating between 40 °C and 4 °C; 6 cycles; RH 75%) was conducted to assess physicochemical stability, ensuring the formulations maintained their quality under temperature fluctuations.

### 2.3. Formulation of Oral Solution Containing DLM, ML, CPC, and Cremophor RH40

This formulation (Table 1) consisted of 0.3% *w*/*w* DLM, 0.2% *w*/*w* ML, 0.05% *w*/*w* CPC, and 1.5% *w*/*w* Cremophor RH40. It was prepared in three parts: the aqueous phase (CPC and glycerin); oil phase 1 (ML, Cremophor RH40, and DLM); and oil phase 2 (Cremophor RH40, menthol, and peppermint oil). The ingredients were mixed at specified temperatures using a hotplate stirrer and homogenizer. After combining, the weight was adjusted to 100 g with distilled water, and a 1% *w*/*w* tartrazine solution was added. The mixture was stirred for 5 min until homogeneous.

### 2.4. Formulation of Nasal Solution Containing DLM, CPC, and Cremophor RH40

This formulation (Table 2) contained 0.2% *w*/*w* DLM, 0.05% *w*/*w* CPC, and 0.75% *w*/*w* Cremophor RH40. The aqueous phase consisted of a CPC solution in a sterile sodium chloride solution, while the oil phase contained DLM, menthol, and Cremophor RH40 in a sodium chloride solution. Both phases were mixed using a homogenizer. The solution was adjusted to 100 g, diluted 20-fold, filtered, and stored in amber glass ampoules under aseptic conditions.

### 2.5. Stability Testing

The samples were stored in amber bottles and subjected to a temperature cycling test, alternating between 40 ± 0.5 °C and 4 ± 0.5 °C for six cycles, with RH maintained at 75 ± 5%. Stability tests evaluated the impact of storage conditions (4 ± 1 °C, 25 ± 1 °C, and 40 ± 1 °C; RH 75 ± 5%) on parameters such as % transmittance, pH, and active ingredients. Microbial contamination was assessed at 1, 3, and 6 months.

### 2.6. Analysis of DLM Using Gas Chromatography–Mass Spectrometry (GC-MS)

The quantity of DLM in nasal and oral formulations was determined using a modified GC-MS analysis method [20].

#### 2.6.1. Preparation of DLM Calibration Curve

Six concentrations (0.25, 0.50, 1.00, 1.50, 2.00, and 2.50% *w*/*w*) were prepared by dissolving DLM in 10% *w*/*w* Tween 20. Five grams of each solution were partitioned with 5 mL of hexane. One milliliter of the hexane solution was placed in a GC vial and then subjected to analysis using a GC-MS instrument. The experiment was performed in triplicate.

#### 2.6.2. Extraction of DLM Samples

Five grams of each sample from the ampoules was partitioned with 5 mL of hexane. One milliliter of the hexane layer was placed in a GC vial and analyzed using a GC-MS instrument. The experiment was performed in triplicate.

#### 2.6.3. GC-MS Analysis

The prepared solution (1.0 μL) was injected into the GC-MS system (Agilent 6890 N, Agilent Technologies, Santa Clara, CA, USA), equipped with electron impact ionization and a mass-selective detector (Agilent 5973, Agilent Technologies, Santa Clara, CA, USA). A DB5-MS column (30 m × 0.25 mm i.d.) was used, with helium as the carrier gas. The temperature program started at 60 °C and increased to 240 °C. Volatile components were identified by comparing the mass spectra with the NIST17 libraries.

### 2.7. Analysis of ML Using Gas Chromatography–Flame Ionization Detector (GC-FID)

A modified GC-FID analysis method [21] was used to determine the quantity of ML in nasal and oral formulations.

#### 2.7.1. Preparation of ML Calibration Curve

The internal standard solution (5 mg/mL) was prepared by dissolving 500 mg of n-tetradecane in 100 mL of pyridine. A 5 mg/mL ML standard solution was made by dissolving 25 mg of ML in 5 mL of the internal standard solution. To construct a standard curve, varying volumes of the stock solution were mixed with BSTFA and TMCS, heated at 70 °C for 30 min, and analyzed using GC-FID.

#### 2.7.2. Extraction of ML Samples

A 20 g sample was extracted by adding 20 mL of dichloromethane in a separating funnel, was shaken for 5 min, and the layers were separated. This process was repeated three times, and the dichloromethane layers were evaporated in a fume hood. The resulting extract was weighed, and a 10 mg/mL stock solution was prepared. The solution was mixed with BSTFA and TMCS, heated at 70 °C for 30 min, cooled, and analyzed using GC-FID.

#### 2.7.3. GC-FID Analysis

The sample was analyzed using a GC-FID system (Agilent Technologies, California, USA) with a DB-5HT column (30 m × 0.25 mm ID and 0.10 µm film). Injection and detector temperatures were set at 350 °C, with helium as the carrier gas at 1.4 mL/min. The column temperature was programmed from 110 °C to 350 °C. The injection volume was 1 µL, with a split ratio of 1:80.

### 2.8. Analysis of CPC Using High-Performance Liquid Chromatography with Diode-Array Detection (HPLC-DAD)

The quantity of CPC in nasal and oral formulations was determined using a modified HPLC-DAD analysis method, as described in reference [22].

#### 2.8.1. Preparation of CPC Calibration Curve

A 1 mg/mL CPC stock solution was prepared by dissolving 5 mg of CPC in acetonitrile, then diluted to 0.5 mg/mL. Standard solutions (18, 24, 30, 36, and 42 µg/mL) were analyzed in duplicate (*n* = 3) using HPLC-DAD.

#### 2.8.2. Extraction of CPC Samples

Nasal and oral formulations (equivalent to 0.05% *w*/*w* CPC) were weighed, dissolved in acetonitrile, and diluted to the final volume with the same solvent to achieve a concentration of 30 µg/mL.

#### 2.8.3. HPLC-DAD Analysis

An HPLC-DAD analysis was performed using an Agilent 1220 Series system (Santa Clara, CA, USA) with an Onyx Monolithic C18 column (25 × 4.6 mm, pore size 2 μm (macropore)/130 Å (mesopore); Phenomenex, Merck KGaA, Darmstadt, Germany). A gradient mobile phase of trifluoroacetic acid (A) and acetonitrile (B) was used. The flow rate was 3.5 mL/min, with a 5 µL injection, 258 nm detection, and a column temperature of 25 °C.

### 2.9. Percentage Transmittance and pH Measurements

The percentage transmittance of the nasal and oral formulations was measured at 660 nm using a UV-Vis spectrophotometer (T60, PG Instrument Limited, Lutterworth, UK), with three replicates per sample. pH was measured at 25 °C using a pH meter (Mettler Toledo SevenEasy, Zürich, Switzerland), also in triplicate.

### 2.10. Contamination

The contamination of heavy metals and microbes in each sample, as obtained in Section 2.6, was determined.

#### 2.10.1. Heavy Metal Contamination

Arsenic (As), cadmium (Cd), lead (Pb), and mercury (Hg) levels were measured to ensure product safety and quality. As reported in our previous study [23], samples were digested using a microwave digester (Model ETHOS ONE, Milestone Corporation, Sorisole, Italy) and analyzed with an inductively coupled plasma mass spectrometer (ICP-MS) (Model 7500ce, Agilent Technologies, Santa Clara, CA, USA). External calibration was performed with an ICP multielement standard solution. All experiments were conducted in triplicate.

#### 2.10.2. Microbial Contamination

A microbial limit test was conducted following USP 43-NF 38 [24] to ensure safety from pathogens. The total aerobic microbial count (TAMC), total combined yeasts/molds count (TYMC), and the presence of *Staphylococcus aureus* and *Pseudomonas aeruginosa* were assessed. Samples were inoculated onto TSA and SDA plates for TAMC and TYMC, respectively, and incubated. Specific microorganisms were detected using selective agar media. Results were recorded as colony-forming units or presence/absence to confirm compliance with USP microbial limits.

### 2.11. In Vitro Antiviral Activity Assay

#### 2.11.1. Anti-SARS-CoV-2 Activity Assays

The anti-SARS-CoV-2 activity of the formulations was evaluated using Vero E6 cells (ATCC CRL-1586™, obtained from the American Type Culture Collection) cultured in MEM supplemented with necessary nutrients. Cytotoxicity was assessed using the MTT assay, and virucidal activity was tested in accordance with ASTM E1053-20 standards [25]. Cell viability was assessed using the MTT assay, where the medium was replaced with 0.5 mg/mL of MTT and incubated at 37 °C for 2 h. After incubation, formazan crystals were dissolved in dimethyl sulfoxide (DMSO), and absorbance was measured at 595 nm using a microplate reader. Cell viability percentages were normalized to the negative control, and the 50% cytotoxic concentration (CC_50_) was calculated. The Delta B.1.617.2 variant of SARS-CoV-2 was isolated from the nasopharyngeal swab of a confirmed COVID-19 patient in Thailand as part of a routine diagnostic procedure at the Tropical Medicine Diagnostic Reference Laboratory, Faculty of Tropical Medicine, Mahidol University. The isolate was authenticated for research use in compliance with institutional and national biosafety regulations. Additionally, permission to use the sample in this study was obtained from the relevant laboratory and institutional authorities (Approval No. MU2023-038, Mahidol University). The isolate had a viral titer of 1 × 10^5^ TCID_50_/mL. The formulations were tested at undiluted concentrations for various contact times. Sodium hypochlorite (0.21% *w*/*v*) was used as a positive control, while MEM served as a negative control. Efficacy was determined based on viral reduction, expressed as percentage efficacy and log reduction values (LRVs). All experiments were performed in triplicate.

Log reduction was calculated using the following formula:Log reduction = log10 (initial viral titer/final viral titer)
where the initial viral titer was measured before treatment, and the final viral titer was determined after exposure to the formulations.

The percentage efficacy was calculated using the following formula:Efficacy (%) = (1 − final viral titer/initial viral titer) × 100

#### 2.11.2. Anti-Influenza Activity Assays

Cytotoxicity of the test compound in Madin-Darby Canine Kidney (MDCK) cells was assessed using the MTT assay. MDCK cells were seeded in 96-well plates, incubated for 24 h, and treated with the test compounds for 10 min. The medium was replaced with 0.5 mg/mL of MTT and incubated for 2 h, followed by the addition of 100 μL of DMSO. Absorbance was then measured at 550 nm, with the 650 nm absorbance subtracted, using a microplate reader. Virucidal activity was assessed following ASTM E1053-20 standards [26]. Test samples were diluted with a MEM medium and evaluated for their ability to inactivate influenza viruses, including FluA(H1N1pdm) (A/Thailand/104/2009), FluA(H3N2) (ATCC VR-1881™), and FluB (ATCC VR-1735™). FluA(H1N1pdm) (A/Thailand/104/2009) was obtained from the National Institute of Health, Thailand, under appropriate institutional agreements. FluA(H3N2) (ATCC VR-1881™) and FluB (ATCC VR-1735™) were sourced from the American Type Culture Collection (ATCC) in compliance with standard biosafety and research protocols. All experiments involving live SARS-CoV-2 and influenza viruses were conducted under strict biosafety conditions at a certified BSL-3 facility at the Faculty of Veterinary Science, Mahidol University. Viral suspensions were mixed with a medium containing bovine serum albumin and dried onto a surface. After exposure to the test products for 30–120 s, the mixtures were neutralized, filtered, and inoculated into cell lines. Cytopathic effects (CPEs) were observed, and viral titers were calculated using the Reed–Muench method [27]. Sodium hypochlorite (0.21% *w*/*v*) served as a positive control [4,28]. Results are reported as log reductions and percentage efficacy, with a 3-log reduction required to demonstrate virucidal activity.

To quantify the reduction in viral load, viral titers were determined using the Reed–Muench method. This method calculates the 50% tissue culture infectious dose (TCID_50_), which is the dilution of virus required to infect 50% of the cells. The formula for TCID_50_ calculation is as follows:TCID_50_ = (*D*_1_ + *D*_2_)/2
where

*D*_1_ = the last dilution at which 100% of the wells show CPE,*D*_2_ = the first dilution at which 0% of the wells show CPE.

This formula calculates the median dilution that causes infection in 50% of the test wells. The result is expressed in units of TCID₅₀ per milliliter, and the viral reduction is calculated by comparing the viral titers before and after exposure to the test formulations.

Units for Viral Titer Calculation: TCID₅₀/mL (50% tissue culture infectious dose per milliliter).

### 2.12. Statistical Analysis

A statistical analysis was performed using a one-way analysis of variance (ANOVA) in the SPSS software version 16, followed by Duncan’s post hoc test at a 95% confidence level (*p* < 0.05). The analysis was based on triplicate measurements, and results with *p*-values less than 0.05 were considered statistically significant.

## 3. Results

### 3.1. Evaluation and Selection of Suitable Surfactants for DLM Formulations

The formulations containing 1% *w*/*w* DLM and 1–15% *w*/*w* surfactant in distilled water were evaluated for physicochemical properties, including % transmittance and pH, one day after preparation and after a temperature cycling test (Table 3; Appendix A). Solutions with DLM and surfactants such as Tween 20, Tween 60, Tween 80, Cremophor RH40, and Cremophor RH60 showed high % transmittance (92.27–98.60%) and suitable pH values (4.64–6.16). High transmittance indicates clear solutions, suggesting effective DLM dissolution. These formulations meet the pH requirements for oral (5.5–7.5) [29] and nasal (4.5–6.5) [30] preparations, making them suitable for antiviral solutions.

Formulations with DLM concentrations above 1% *w*/*w* (2–5% *w*/*w*) and 10% *w*/*w* Tween 20 were unstable, showing turbidity and phase separation after preparation and stability testing. The DLM-to-Tween 20 ratio of 1:10 was more suitable. Formulations with coco glucoside and decyl glucoside exhibited low % transmittance and high pH values (Table 3). Similarly, Poloxamer 184 and Poloxamer 407 at DLM-to-surfactant ratios of 1:5 to 1:10 produced turbid solutions before and after the temperature cycling test. Due to these issues, these surfactants were not selected for developing DLM antiviral solutions.

Tween 20 is suitable for oral solutions due to its low molecular weight, high hydrophilicity, mild taste, and proven safety [31,32]. However, Cremophor RH40 is better for oral and nasal formulations due to its lower hydrophilic–lipophilic balance (HLB), enhancing DLM solubilization and stability in aqueous solutions [33]. The DLM and Cremophor RH40 formulation (1:5 ratio) also has a milder bitter taste than the DLM and Tween 20 formulation (1:10 ratio), improving patient compliance. Based on experimental data (Table 3), Cremophor RH40 was chosen for further development due to its superior solubilization and taste profile.

### 3.2. Development of Formulations Containing DLM, CPC, ML, and Cremophor RH40

Formulations with Tween 20, Tween 60, or Tween 80 had undesirable bitter and soapy tastes. Cremophor RH40, with a milder taste, emerged as more suitable for oral use. The DLM–Cremophor RH40 formulation (1:5 ratio) showed excellent % transmittance (96.90–94.20%) and appropriate pH (6.05–5.98), making it the preferred choice for oral formulations (Table 3).

The study showed that using Cremophor RH40 at five times the DLM concentration ensured stability. Formulations with reduced DLM (0.3–0.5% *w*/*w*) and added ML (0.1–0.5% *w*/*w*) or CPC (0.05–0.075% *w*/*w*) maintained anti-SARS-CoV-2 efficacy, comparable to 1% *w*/*w* DLM alone. Formulation D (Figure 1), containing 0.3% DLM, 0.05% CPC, 0.2% ML, and 1.5% Cremophor RH40, exhibited 99.99% virucidal activity against SARS-CoV-2 (log reduction 3.8750) in 120 s (Table 4), making it suitable for mouth spray or mouthwash applications.

Cremophor RH40 was chosen for its mildness and low irritation potential in nasal formulations, with safety at 0.75% *w*/*w* [34,35]. Formulation E (Figure 1), containing 0.2% DLM, 0.05% CPC, and 0.75% Cremophor RH40, demonstrated 99.87% virucidal activity against SARS-CoV-2 (log reduction 2.9063) within 120 s (Table 4), making it suitable as a nasal spray or rinse.

### 3.3. Virucidal Activity and Cytotoxicity of Formulations Containing DLM, CPC, ML, and Cremophor RH40

The Appendix A provide raw data, including cytotoxicity and neutralization validation control data for oral formulation D and nasal formulation E in MDCK and Vero cells. These tables also present cytotoxicity test results for both formulations at a 1:32 dilution against FluA(H1N1pdm) (A/Thailand/104/2009), FluA(H3N2) (ATCC VR-1881™), FluB (ATCC VR-1735™), and SARS-CoV-2. Additionally, virus recovery data for these viral strains are included, along with virucidal activity data for oral formulation D and nasal formulation E at a 1:32 dilution against FluA(H1N1pdm) (A/Thailand/104/2009), FluA(H3N2) (ATCC VR-1881™), FluB (ATCC VR-1735™), and SARS-CoV-2.

Cytotoxicity testing of DLM on Vero E6 cells (CRL-1586™) showed 99.79% viability at 0.125% and 81.81% at 0.25%. Formulations for anti-SARS-CoV-2 activity were based on concentrations with >80% cell viability. As shown in Table 4, formulation A (1% DLM and 1.5% Cremophor RH40) achieved 99.93–99.98% viral reduction (log reductions 3.1875–3.8125) within 30 s to 10 min. Formulation B (0.5% DLM, 0.5% ML, and 1.5% Cremophor RH40) showed 99.97% efficacy (log reduction 3.6042) at 10 min. Formulation C (0.3% DLM, 0.05% CPC, and 1.5% Cremophor RH40) achieved 99.92% efficacy (log reduction 3.0833) within 120 s, demonstrating strong virucidal activity across all tested formulations. Formulation D (0.3% DLM, 0.05% CPC, 0.2% ML, and 1.5% Cremophor RH40) achieved the highest virucidal efficacy, with a 3.875 log reduction and 99.99% efficacy at 120 s, making it ideal for mouth spray or mouthwash applications. Formulation E (0.2% DLM, 0.05% CPC, and 0.75% Cremophor RH40) showed a 2.906 log reduction and 99.87% efficacy, suitable for nasal sprays or rinses due to its milder composition. The positive control (0.21% sodium hypochlorite) demonstrated a >4.4 log reduction and 99.99% efficacy. A statistical analysis (*p* < 0.05) revealed significant differences in log reductions among formulations, with formulations A and D performing similarly. However, all formulations showed a comparable percent efficacy, indicating consistent antiviral effectiveness across different compositions.

As shown in Table 5, formulations D (oral) and E (nasal) exhibited >99.99% anti-influenza efficacy against FluA(H1N1), FluA(H3N2), and FluB, with log reductions >4.000 across all concentrations, dilutions (1:2–1:32), and contact times (30–120 s). Tested in quadruplicate, both formulations demonstrated consistent, non-dose-dependent antiviral activity, highlighting their potential as robust anti-influenza agents.

Moreover, the formulations effectively protected Vero E6 and MDCK cells from the cytopathic effects (CPEs) induced by SARS-CoV-2 and influenza viruses, respectively. The treated cells maintained their normal morphology and viability, indicating that the formulations not only inactivated the viruses but also prevented virus-induced cellular damage. These findings further support the potential application of these formulations in antiviral treatments.

### 3.4. Assessment of Heavy Metals and Microbial Contamination in Oral and Nasal Formulations Containing DLM, CPC, ML, and Cremophor RH40

Formulations D (oral) and E (nasal) met USP 2024 safety standards [36], with undetectable heavy metals (below LOQ: As < 0.041 ppm, Cd < 0.021 ppm, Pb < 0.020 ppm, and Hg < 0.054 ppm) and microbial loads (TAMC < 10 cfu/mL and TYMC < 10 cfu/mL). *S. aureus* and *P. aeruginosa* were absent, confirming compliance with pharmaceutical safety criteria for oral and nasal use [37].

### 3.5. An Evaluation of the Stability of Oral and Nasal Formulations Containing DLM, CPC, ML, and Cremophor RH40

DLM in oral formulation D and nasal formulation E was quantified using GC-MS, with a linear calibration curve (0.25–2.5% *w*/*w*) and an R^2^ of 0.9978. LOD and LOQ were 0.017% *w*/*w* and 0.042% *w*/*w*, respectively. CPC was measured in both formulations using HPLC-DAD, with a linear range of 18–42 µg/mL and an R^2^ of 0.9992. LOD and LOQ were 1.01 µg/mL and 3.05 µg/mL, respectively. ML in oral formulation D was quantified using GC-FID, with an R^2^ of 0.9998, a LOD of 0.072 mg/mL, and an LOQ of 0.217 mg/mL. All methods showed high precision and accuracy.

The stability study of oral formulation D and nasal formulation E under various storage conditions showed notable trends in % transmittance; pH; and the levels of DLM, CPC, and ML (Table 6). Oral formulation D maintained relatively stable transmittance at 4 °C for 3 months, with a decline at higher temperatures. DLM showed significant degradation at 40 °C, while CPC remained stable. ML decreased at higher temperatures. Despite these changes, the formulations met microbiological standards, conforming to USP 2024 acceptance criteria [37].

Nasal formulation E showed stable transmittance at 4 °C, with a slight increase over 6 months, while a decline was observed at 25 °C and a significant drop at 40 °C. DLM retention was high at lower temperatures but decreased at 40 °C. CPC remained stable, and the pH showed minimal changes. All samples met microbiological safety standards, conforming to USP 2024 criteria [37].

## 4. Discussion

Oral formulation D and nasal formulation E were developed to optimize taste, stability, and safety. Cremophor RH40 was selected for its mild taste and low irritation potential [34]. Formulation D, for oral use, demonstrated strong antiviral activity against SARS-CoV-2 and influenza, with high efficacy (99.99%), while formulation E, for nasal use, exhibited effective antiviral activity, although it was slightly lower for SARS-CoV-2, and was highly effective against influenza. Both formulations are considered safe and effective, pending clinical validation, with formulation D being more suitable for oral applications and formulation E for nasal use.

Despite these promising results, several limitations and potential sources of bias should be considered. First, the in vitro nature of this study means the results may not fully reflect the clinical efficacy or safety profile of the formulations in humans. Additionally, while blind coding was employed to reduce experimental bias, there could still be unforeseen biases in formulation preparation or testing. Furthermore, the formulations were tested under specific storage conditions, and while they demonstrated stability at 4 °C, higher temperatures, especially at 40 °C, led to significant degradation of DLM and ML. These findings may not fully account for the wide range of temperature variations encountered during real-world storage and transport. Therefore, future studies should investigate these formulations under diverse temperature conditions and explore longer-term stability to better simulate real-life storage and use. Another potential limitation is the exclusion of ML from formulation E due to its thermal sensitivity; this decision, while improving formulation stability, could also impact the antiviral efficacy of formulation E in certain environments. Finally, while Cremophor RH40 contributed to the stability of both formulations, further research is needed to evaluate its long-term safety and any potential interactions with other excipients or active ingredients in the formulations. These factors highlight the need for careful consideration of storage conditions, formulation components, and further in vivo testing before clinical application.

## 5. Conclusions

Formulation D (0.3% *w*/*w* DLM, 0.05% *w*/*w* CPC, 0.2% *w*/*w* ML, and 1.5% *w*/*w* Cremophor RH40) demonstrated exceptional antiviral performance, achieving 99.99% efficacy against SARS-CoV-2 within 120 s, making it ideal for oral use. Formulation E (0.2% *w*/*w* DLM, 0.05% *w*/*w* CPC, and 0.75% *w*/*w* Cremophor RH40) showed 99.87% efficacy and is suitable for nasal applications. Both formulations also exhibited strong efficacy against influenza viruses, maintaining >99.99% efficacy across various concentrations and contact times. Stability testing confirmed minimal changes in active compounds and no microbial contamination at 4 °C. These formulations hold significant promise for clinical use in preventing and managing viral infections, particularly in healthcare settings. Clinical trials evaluating their safety and efficacy in COVID-19 patients are underway, with the goal of advancing global health.

## Figures and Tables

**Figure 1 pharmaceutics-17-00349-f001:**
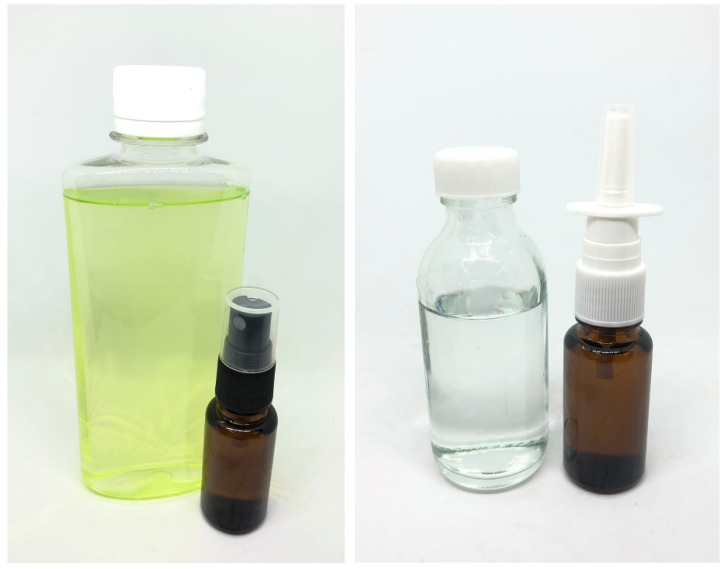
Physical characteristics of oral formulation D (**left**) and nasal formulation E (**right**).

**Table 1 pharmaceutics-17-00349-t001:** Formulation of oral solution containing DLM, ML, CPC, and Cremophor RH40.

Ingredient	Concentration (% *w*/*w*)	Part
DLM	0.3	Oil part 1
ML	0.2	
Cremophor RH40	0.5	
Menthol	0.1	Oil part 2
Peppermint oil	0.1	
Cremophor RH40	1.0	
CPC	0.05	Aqueous part
Glycerin	25	
1% *w*/*w* Tartrazine (INS No. 102) aqueous solution	1	
Distilled water	q.s. solution to 100 g	

**Table 2 pharmaceutics-17-00349-t002:** Formulation of nasal solution containing DLM, CPC, and Cremophor RH40.

Ingredient	Concentration (% *w*/*w*)		Part
	Before Dilution	After a 1:20 Dilution	
DLM	4.20	0.20	Oil part
Cremophor RH40	15.75	0.75	
Menthol	0.21	0.01	
CPC	1.05	0.05	Aqueous part
Sterile 0.9% *w/v* sodium chloride solution	q.s. solution to 100 g	q.s. solution to 1050 g	

**Table 3 pharmaceutics-17-00349-t003:** % transmittance and pH values of formulations containing DLM and surfactants at different ratios that were measured 1 day after preparation and after a temperature cycling test (6 cycles).

Surfactant	Ratio of DLM to Surfactant	% Transmittance		pH	
		1 Day	Temperature Cycling Test	1 Day	Temperature Cycling Test
Tween 20	1:5	0.10 ± 0.00 ^Z^	0.83 ± 0.01 ^X,Y^	6.40 ± 0.08 ^c^	5.98 ± 0.12 ^d,e^
	1:6	0.30 ± 0.00 ^Y,Z^	0.77 ± 0.00 ^Y^	5.82 ± 0.12 ^d,e,f^	5.40 ± 0.09 ^g^
	1:7	1.63 ± 0.06 ^X^	1.13 ± 0.12 ^X^	5.83 ± 0.14 ^d,e,f^	5.39 ± 0.26 ^g,h^
	1:8	4.90 ± 0.36 ^V^	52.40 ± 0.33 ^P^	6.20 ± 0.02 ^c,d^	5.47 ± 0.22 ^g^
	1:9	94.63 ± 0.06 ^F^	81.10 ± 0.00 ^L^	5.90 ± 0.09 ^d,e^	5.41 ± 0.17 ^g^
	1:10	97.70 ± 0.00 ^B^	92.27 ± 0.05 ^G,H^	5.80 ± 0.10 ^d,e,f^	5.75 ± 0.07 ^e,f^
Tween 60	1:2.5	31.10 ± 0.10 ^R,S^	25.23 ± 0.29 ^T^	5.63 ± 0.07 ^e,f,g^	4.98 ± 0.21 ^h^
	1:5	65.50 ± 0.00 ^N^	60.90 ± 0.08 ^O^	5.14 ± 0.21 ^g,h^	5.02 ± 0.05 ^h^
	1:10	91.67 ± 0.06 ^H^	95.77 ± 0.05 ^D^	4.91 ± 0.11 ^h^	4.84 ± 0.09 ^h^
	1:15	98.13 ± 0.06 ^A^	98.60 ± 0.00 ^A^	4.92 ± 0.11 ^h^	4.64 ± 0.12 ^i^
Tween 80	1:5	0.20 ± 0.00 ^Z^	0.57 ± 0.05 ^Y,Z^	6.53 ± 0.00 ^c^	6.08 ± 0.17 ^d,e^
	1:10	20.20 ± 0.00 ^T,U^	14.90 ± 0.08 ^U^	6.23 ± 0.05 ^c,d^	6.18 ± 0.05 ^c,d^
	1:12.5	94.50 ± 0.00 ^F^	98.50 ± 0.00 ^A^	6.41 ± 0.12 ^c^	5.59 ± 0.21 ^f,g^
	1:15	98.23 ± 0.12 ^A^	96.80 ± 0.00 ^C^	6.04 ± 0.21 ^d,e^	5.96 ± 0.06 ^d,e^
Cremophor RH40	1:1	10.47 ± 0.46 ^U^	10.30 ± 0.08 ^U^	6.46 ± 0.21 ^c^	6.01 ± 0.07 ^d,e^
	1:2.5	90.30 ± 0.00 ^I^	90.23 ± 0.21 ^I^	5.98 ± 0.11 ^d,e^	5.73 ±0.11 ^e,f^
	1:5	96.90 ± 0.00 ^C^	94.20 ± 0.00 ^F^	6.05 ± 0.11 ^d,e^	5.98 ± 0.09 ^d,e^
	1:10	95.10 ± 0.00 ^E^	93.93 ± 0.12 ^G^	5.82 ± 0.12 ^d,e,f^	5.59 ± 0.16 ^f,g^
Cremophor RH60	1:1	0.60 ± 0.00 ^Y,Z^	0.80 ± 0.08 ^X,Y^	6.40 ± 0.07 ^c^	6.17 ± 0.21 ^c,d^
	1:2.5	42.12 ± 0.10 ^Q^	37.20 ± 0.16 ^R^	5.78 ± 0.08 ^e,f^	5.79 ± 0.12 ^e,f^
	1:5	95.90 ± 0.00 ^D^	97.57 ± 3.63 ^B^	6.16 ± 0.17 ^c,d^	5.95 ± 0.15 ^d,e^
	1:10	97.00 ± 0.00 ^C^	91.10 ± 0.00 ^H^	5.83 ± 0.12 ^d,e,f^	5.97 ± 0.07 ^d,e^
Coco glucoside	1:2.5	24.40 ± 0.00 ^T^	21.20 ± 0.00 ^T^	10.94 ± 0.04 ^b^	10.53 ± 0.17 ^b^
	1:5	80.83 ± 0.06 ^L^	74.50 ± 0.08 ^M^	11.26 ± 0.14 ^a,b^	11.17 ± 0.05 ^a,b^
	1:7.5	88.10 ± 0.00 ^J^	81.97 ± 0.09 ^K,L^	11.40 ± 0.07 ^a^	11.18 ± 0.09 ^a,b^
	1:10	82.73 ± 0.06 ^K^	82.70 ± 0.08 ^K^	11.57 ± 0.06 ^a^	11.46 ± 0.05 ^a^
Decyl glucoside	1:2.5	6.60 ± 0.00 ^V^	4.20 ± 0.00 ^V^	10.85 ± 0.11 ^b^	10.70 ± 0.12 ^b^
	1:5	30.20 ± 0.00 ^R,S^	28.83 ± 0.12 ^T^	11.14 ± 0.05 ^a,b^	11.01 ±0.06 ^a,b^
	1:7.5	94.83 ± 0.06 ^E,F^	95.80 ± 0.00 ^D^	11.23 ± 0.07 ^a,b^	11.10 ± 0.17 ^a,b^
	1:10	92.90 ± 0.00 ^G^	94.80 ± 0.00 ^E,F^	11.36 ± 0.03 ^a^	11.30 ± 0.07 ^a^

The analysis results are expressed as means ± standard deviation (mean ± SD). Difference between capital letters in % transmittance and between a–i in pH values represent significant differences at a 95% confidence interval (*p* < 0.05).

**Table 4 pharmaceutics-17-00349-t004:** Anti-SARS-CoV-2 activity of formulations containing DLM, CPC, and ML with Cremophor RH40 as the surfactant.

Formulations	Ingredients	Contact Time	Log Reduction	Statistical Results	% Efficacy	Statistical Results
A	1% *w*/*w* DLM 1.5% *w*/*w* Cremophor RH40	30 s	3.1875 ± 0.0722	c	99.93 ± 0.0108	a
		1 min	3.2708 ± 0.0722		99.95 ± 0.0089	
		5 min	3.6875 ± 0.1382		99.98 ± 0.0063	
		10 min	3.8125 ± 0.0722		99.98 ± 0.0026	
B	0.5% *w*/*w* DLM 0.5% *w*/*w* ML 1.5% *w*/*w* Cremophor RH40	30 s	2.9063 ± 0.0625	bc	99.87 ± 0.0167	a
		1 min	3.0625 ± 0.0722		99.91 ± 0.0144	
		5 min	3.2500 ± 0.1021		99.94 ± 0.0138	
		10 min	3.6042 ± 0.0722		99.97 ± 0.0041	
C	0.3% *w*/*w* DLM 0.05% *w*/*w* CPC 1.5% *w*/*w* Cremophor RH40	30 s	2.1875 ± 0.0722	ab	99.34 ± 0.1083	a
		60 s	2.7604 ± 0.0859		99.82 ± 0.0336	
		90 s	2.9375 ± 0.0722		99.88 ± 0.0193	
		120 s	3.0833 ± 0.1021		99.92 ± 0.019	
D	0.3% *w*/*w* DLM 0.05% *w*/*w* CPC 0.2% *w*/*w* ML 1.5% *w*/*w* Cremophor RH40 Other excipients as shown in Table 3	30 s	3.2083 ± 0.0589	c	99.94 ± 0.0088	a
		60 s	3.2708 ± 0.0722		99.95 ± 0.0089	
		90 s	3.6250 ± 0.1021		99.98 ± 0.0055	
		120 s	3.8750 ± 0.1021		99.99 ± 0.0032	
E	0.2% *w*/*w* DLM 0.05% *w*/*w* CPC 0.75% *w*/*w* Cremophor RH40 Other excipients as shown in Table 4	30 s	2.2708 ± 0.0722	a	99.46 ± 0.0894	a
		60 s	2.6042 ± 0.0722		99.75 ± 0.0415	
		90 s	2.7917 ± 0.1021		99.84 ± 0.0396	
		120 s	2.9063 ± 0.1197		99.87 ± 0.0369	

Statistical significance was evaluated using ANOVA followed by Duncan’s post hoc test at a 95% confidence level (*p* < 0.05). Different letters denote a significant difference.

**Table 5 pharmaceutics-17-00349-t005:** Anti-influenza activity of formulations containing DLM, CPC, and ML with Cremophor RH40 as the surfactant.

Formulation	Concentration	Contact Time (s)	FluA(H1N1)		FluA(H3N2)		FluB	
			Efficacy *	Log Reduction *	Efficacy *	Log Reduction *	Efficacy *	Log Reduction *
**D**	Conc	30	>99.99%	>4.000	>99.99%	>4.000	>99.99%	>4.000
	1:2		>99.99%	>4.000	>99.99%	>4.000	>99.99%	>4.000
	1:4		>99.99%	>4.000	>99.99%	>4.000	>99.99%	>4.000
	1:8		>99.99%	>4.000	>99.99%	>4.000	>99.99%	>4.000
	1:16		>99.99%	>4.000	>99.99%	>4.000	>99.99%	>4.000
	1:32		>99.99%	>4.000	>99.99%	>4.000	>99.99%	>4.000
	Conc	60	>99.99%	>4.000	>99.99%	>4.000	>99.99%	>4.000
	1:2		>99.99%	>4.000	>99.99%	>4.000	>99.99%	>4.000
	1:4		>99.99%	>4.000	>99.99%	>4.000	>99.99%	>4.000
	1:8		>99.99%	>4.000	>99.99%	>4.000	>99.99%	>4.000
	1:16		>99.99%	>4.000	>99.99%	>4.000	>99.99%	>4.000
	1:32		>99.99%	>4.000	>99.99%	>4.000	>99.99%	>4.000
	Conc	90	>99.99%	>4.000	>99.99%	>4.000	>99.99%	>4.000
	1:2		>99.99%	>4.000	>99.99%	>4.000	>99.99%	>4.000
	1:4		>99.99%	>4.000	>99.99%	>4.000	>99.99%	>4.000
	1:8		>99.99%	>4.000	>99.99%	>4.000	>99.99%	>4.000
	1:16		>99.99%	>4.000	>99.99%	>4.000	>99.99%	>4.000
	1:32		>99.99%	>4.000	>99.99%	>4.000	>99.99%	>4.000
	Conc	120	>99.99%	>4.000	>99.99%	>4.000	>99.99%	>4.000
	1:2		>99.99%	>4.000	>99.99%	>4.000	>99.99%	>4.000
	1:4		>99.99%	>4.000	>99.99%	>4.000	>99.99%	>4.000
	1:8		>99.99%	>4.000	>99.99%	>4.000	>99.99%	>4.000
	1:16		>99.99%	>4.000	>99.99%	>4.000	>99.99%	>4.000
	1:32		>99.99%	>4.000	>99.99%	>4.000	>99.99%	>4.000
**E**	Conc	30	>99.99%	>4.000	>99.99%	>4.000	>99.99%	>4.000
	1:2		>99.99%	>4.000	>99.99%	>4.000	>99.99%	>4.000
	1:4		>99.99%	>4.000	>99.99%	>4.000	>99.99%	>4.000
	1:8		>99.99%	>4.000	>99.99%	>4.000	>99.99%	>4.000
	1:16		>99.99%	>4.000	>99.99%	>4.000	>99.99%	>4.000
	1:32		>99.99%	>4.000	>99.99%	>4.000	>99.99%	>4.000
	Conc	60	>99.99%	>4.000	>99.99%	>4.000	>99.99%	>4.000
	1:2		>99.99%	>4.000	>99.99%	>4.000	>99.99%	>4.000
	1:4		>99.99%	>4.000	>99.99%	>4.000	>99.99%	>4.000
	1:8		>99.99%	>4.000	>99.99%	>4.000	>99.99%	>4.000
	1:16		>99.99%	>4.000	>99.99%	>4.000	>99.99%	>4.000
	1:32		>99.99%	>4.000	>99.99%	>4.000	>99.99%	>4.000
	Conc	90	>99.99%	>4.000	>99.99%	>4.000	>99.99%	>4.000
	1:2		>99.99%	>4.000	>99.99%	>4.000	>99.99%	>4.000
	1:4		>99.99%	>4.000	>99.99%	>4.000	>99.99%	>4.000
	1:8		>99.99%	>4.000	>99.99%	>4.000	>99.99%	>4.000
	1:16		>99.99%	>4.000	>99.99%	>4.000	>99.99%	>4.000
	1:32		>99.99%	>4.000	>99.99%	>4.000	>99.99%	>4.000
	Conc	120	>99.99%	>4.000	>99.99%	>4.000	>99.99%	>4.000
	1:2		>99.99%	>4.000	>99.99%	>4.000	>99.99%	>4.000
	1:4		>99.99%	>4.000	>99.99%	>4.000	>99.99%	>4.000
	1:8		>99.99%	>4.000	>99.99%	>4.000	>99.99%	>4.000
	1:16		>99.99%	>4.000	>99.99%	>4.000	>99.99%	>4.000
	1:32		>99.99%	>4.000	>99.99%	>4.000	>99.99%	>4.000

* Results are based on quadruplicate experiments.

**Table 6 pharmaceutics-17-00349-t006:** % transmittance; pH values; and % label amounts of DLM, ML, and CPC in oral formulation D and nasal formulation E stored at 4 ± 1 °C, 25 ± 1 °C, and 40 ± 1 °C for 1, 3, and 6 months.

Storage Temperature	Storage Time	% Transmittance	pH	% Label Amount			Microbial Contamination
				DLM	CPC	ML	
Oral formulation D							
Fresh prepared		71.37 ± 0.05 ^c^	6.05 ± 0.01 ^g^	96.95 ± 0.01 ^g^	102.06 ± 0.06 ^f^	87.54 ± 0.05 ^i^	Conform
4 ± 1 °C	1 month	89.50 ± 0.08 ^h^	6.03 ± 0.01 ^g^	97.05 ± 0.12 ^g^	102.76 ± 0.12 ^g^	76.05 ± 0.04 ^h^	Conform
	3 months	89.53 ± 0.05 ^h^	6.01 ± 0.01 ^f^	96.98 ± 0.11 ^g^	102.89 ± 0.10 ^g^	74.91 ± 0.18 ^g^	Conform
	6 months	74.27 ± 0.02 ^d^	5.96 ± 0.00 ^d^	95.32 ± 0.05 ^f^	101.96 ± 0.06 ^f^	68.04 ± 0.30 ^d^	Conform
25 ± 1 °C	1 month	81.20 ± 0.05 ^g^	5.98 ± 0.01 ^e^	95.34 ± 0.19 ^f^	101.99 ± 0.29 ^f^	75.21 ± 0.01 ^g^	Conform
	3 months	81.20 ± 0.70 ^g^	5.93 ± 0.02 ^c^	94.12 ± 0.05 ^e^	98.89 ± 0.07 ^d^	74.21 ± 0.07 ^f^	Conform
	6 months	76.10 ± 0.00 ^e^	5.91 ± 0.03 ^b^	89.14 ± 0.16 ^d^	95.94 ± 0.18 ^c^	66.40 ± 0.21 ^c^	Conform
40 ± 1 °C	1 month	69.07 ± 0.10 ^b^	5.96 ± 0.01 ^d^	87.45 ± 0.25 ^c^	99.29 ± 0.11 ^e^	72.19 ± 0.45 ^e^	Conform
	3 months	79.30 ± 0.00 ^f^	5.90 ± 0.02 ^b^	78.95 ± 0.11 ^b^	95.59 ± 0.07 ^b^	60.02 ± 0.15 ^b^	Conform
	6 months	63.87 ± 0.06 ^a^	5.85 ± 0.02 ^a^	66.87 ± 0.20 ^a^	93.06 ± 0.16 ^a^	57.67 ± 0.02 ^a^	Conform
Nasal formulation E							
Freshly prepared		88.53 ± 0.07 ^c^	4.78 ± 0.01 ^c^	97.57 ± 0.03 ^i^	102.12 ± 0.10 ^g^	NA	Conform
4 ± 1 °C	1 month	90.23 ± 0.13 ^e^	4.76 ± 0.01 ^bc^	97.32 ± 0.11 ^hi^	102.09 ± 0.08 ^g^	NA	Conform
	3 months	90.43 ± 0.06 ^e^	4.75 ± 0.02 ^ab^	97.08 ± 0.13 ^gh^	102.10 ± 0.17 ^g^	NA	Conform
	6 months	90.34 ± 0.12 ^e^	4.75 ± 0.01 ^ab^	96.56 ± 0.17 ^ef^	101.56 ± 0.09 ^f^	NA	Conform
25 ± 1 °C	1 month	89.45 ± 0.10 ^d^	4.75 ± 0.01 ^ab^	96.77 ± 0.33 ^fg^	102.03 ± 0.14 ^g^	NA	Conform
	3 months	87.78 ± 0.10 ^b^	4.75 ± 0.01 ^ab^	96.19 ± 0.04 ^de^	100.04 ± 0.51 ^d^	NA	Conform
	6 months	87.67 ± 0.40 ^b^	4.74 ± 0.02 ^ab^	95.86 ± 0.13 ^d^	98.05 ± 0.32 ^c^	NA	Conform
40 ± 1 °C	1 month	80.09 ± 0.04 ^a^	4.75 ± 0.01 ^ab^	92.89 ± 0.13 ^c^	100.59 ± 0.03 ^e^	NA	Conform
	3 months	80.10 ± 0.22 ^a^	4.74 ± 0.01 ^ab^	85.99 ± 0.55 ^b^	96.08 ± 0.09 ^b^	NA	Conform
	6 months	79.87 ± 0.52 ^a^	4.73 ± 0.02 ^a^	80.43 ± 0.13 ^a^	94.11 ± 0.22 ^a^	NA	Conform

NA: not analyzed. Different letters above the mean ± SD values indicate a statistically significant difference at *p* < 0.05.

## Data Availability

The data supporting the findings of this study are available from the corresponding author, Chutima Limmatvapirat, upon reasonable request.

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
