# Peer review of "In Vitro Development of Local Antiviral Formulations with Potent Virucidal Activity Against SARS-CoV-2 and Influenza Viruses"

_pharmaceutics, 2025, doi:10.3390/pharmaceutics17030349_

Round 1
Reviewer 1 Report
Comments and Suggestions for Authors
The title is not accurate "Development of Innovative Antiviral Formulations with Potent Virucidal Activity Against SARS-CoV-2 and Influenza Viruses"
The effect mentioned in the study is local. Should be added in the title
The raw data must be more precisely described: CPE of both viruses H1N1, H3N2, SARSCov2, especially (with figures of the CPE)
Why the authors did not test dilutions above 1:32?
It would relevant to test H5N1 strains
This work is not very interesting with this current version
Comments on the Quality of English LanguageCorrect
Reviewer 2 Report
Comments and Suggestions for Authors
1. My strong recommendation is to revise the title and the Abstract by inserting the term “in vitro”, since it creates a misconception that these formulations help treat the disease in vivo, although the authors only tested various compositions against purified viral particles. Otherwise, the title and abstract become misleading.
2. Fig. 1 should be removed from the main manuscript, and put into the Supplements, since it does not contain any necessary information for the main results.
3. On the contary, the manuscript lacks many important experimental details and needs thorough revision. Thus, in the methods, please provide details, including formulas for how a log reduction and efficacy were calculated.
4. The MTT test is indicated in the methods, but the raw data for this test are not provided. Please provide the data for this test, as is usually to provide them with absorption indices at a wavelength.
5. No data are provided for Cytopathic effects when assessing Virucidal activity. Please provide details of the method, in what units they were measured, what the formula was. Are there any raw measurement data that convincingly indicate a virucidal effect, photographs, etc.? The authors stated that "Viral suspensions were mixed with medium containing bovine serum albumin and dried onto a surface. After exposure to the test products for 30–120 sec, the mixtures were neutralized, filtered, and inoculated into cell lines.»
6. Were there any negative controls when assessing Virucidal activity, please provide data, photographs, etc.
7. It is not stated whether Blind coding was used when setting up the experiment and analyzing the samples to avoid any research biases. Otherwise it could unconsciously influence the authors’ experiment's results based on the authors’ expectations or preferences.
8. Please expand the Discussion section to discuss possible limitations and biases that could have influenced the experimental results.
Round 2
Reviewer 1 Report
Comments and Suggestions for Authors
Under these revision conditions the paper is acceptable
Reviewer 2 Report
Comments and Suggestions for Authors
The authors successfully address all my comments, therefore I do not have any further questions.